# The Cys Sense: Thiol Redox Switches Mediate Life Cycles of Cellular Proteins

**DOI:** 10.3390/biom11030469

**Published:** 2021-03-22

**Authors:** Meytal Radzinski, Tal Oppenheim, Norman Metanis, Dana Reichmann

**Affiliations:** 1Department of Biological Chemistry, The Alexander Silberman Institute of Life Sciences, Safra Campus Givat Ram, The Hebrew University of Jerusalem, Jerusalem 91904, Israel; meytal.radzinski@mail.huji.ac.il (M.R.); tal.oppenheim1@mail.huji.ac.il (T.O.); 2Institute of Chemistry, Safra Campus Givat Ram, The Hebrew University of Jerusalem, Jerusalem 91904, Israel; metanis@mail.huji.ac.il

**Keywords:** thiol switches, proteostasis, chaperones, protein degradation, oxidative stress, redox-regulated proteins

## Abstract

Protein homeostasis is an essential component of proper cellular function; however, sustaining protein health is a challenging task, especially during the aerobic lifestyle. Natural cellular oxidants may be involved in cell signaling and antibacterial defense; however, imbalanced levels can lead to protein misfolding, cell damage, and death. This merges together the processes of protein homeostasis and redox regulation. At the heart of this process are redox-regulated proteins or thiol-based switches, which carefully mediate various steps of protein homeostasis across folding, localization, quality control, and degradation pathways. In this review, we discuss the “redox code” of the proteostasis network, which shapes protein health during cell growth and aging. We describe the sources and types of thiol modifications and elaborate on diverse strategies of evolving antioxidant proteins in proteostasis networks during oxidative stress conditions. We also highlight the involvement of cysteines in protein degradation across varying levels, showcasing the importance of cysteine thiols in proteostasis at large. The individual examples and mechanisms raised open the door for extensive future research exploring the interplay between the redox and protein homeostasis systems. Understanding this interplay will enable us to re-write the redox code of cells and use it for biotechnological and therapeutic purposes.

## 1. Introduction

The stability of the cellular proteome is constantly challenged by conditions that cause proteotoxic stress, including errors during protein synthesis, undesirable protein modification (e.g., oxidation), inherited polymorphisms, and native changes in physiological conditions, such as aging [1,2,3]. It is, therefore, not surprising that protein homeostasis (or proteostasis) is among the most important mechanisms maintaining the proper balance between protein biogenesis and its cellular function. Thus, proteostasis can be seen as a pivotal player in maintaining the functional proteome specifically and cell survival more broadly.

The delicate balance in proteostasis is achieved by a carefully and timely regulated protein network (i.e., the proteostasis network), which is composed of a high number of proteins carrying out different functions (folding, protein editing, transfer, and degradation) across different cellular organelles, including the cytosol [4], nucleus [5], mitochondria [6,7], and others. This protein network is adaptive and comprises thousands of proteins in high eukaryotes [4,8], some of which are redundant in their function and some specific to either cellular condition [9,10] or client protein [11]. Despite the overall high energy cost in protein production within the cell, proteins mediating protein folding and degradation comprise a large portion of its proteome, emphasizing the significant role of this system in cell functionality. The composition and dynamics of the proteostasis network are shaped by various changes in cellular requirements and conditions, (e.g., the accumulation of oxidants, environmental temperatures such as heat shock, and pH), each of which may challenge cellular homeostasis and protein stability. Stress response mechanisms heavily rely on adaptation of the proteostasis system often in a near-instantaneous manner [12,13] in order to ensure temporal control of “health” and the functionality of thousands of cellular proteins. While some of these adaptive processes may involve relatively few proteins, others may lead to a significant rearrangement of the cellular proteome and formation of stress bodies, particularly in combination with other stress factors [14,15,16,17].

Proteostasis defines and carefully synchronizes every step from a protein’s initial formation by the ribosome through trafficking to subcellular compartments under appropriate conditions, assembly to sophisticated and dynamic macro-complexes, protein modification, functional regulation, signaling, and finally its “death” and turnover. A breakdown of this process can lead to the formation of misfolded proteins and potential accumulation of toxic protein aggregates, damaging cell growth and activity [14,15,18].

The heart of the protein proteostasis system is, thus, chaperones and co-chaperones, which are responsible for recognizing aggregation-prone proteins and assisting in their proper folding [4,19,20]. Chaperones range in function and behavior from the broadly conserved and highly prevalent (e.g., Hsp40/DnaJ, Hsp70/DnaK, etc.) [4] to the client- or condition-specific proteins (e.g., redox-regulated Hsp33 [21], pH-regulated HdeA [22] and HdeB [23], heat-induced small heat shock proteins (sHSPs) [24,25,26]). The chaperones differ not only in their mechanistic functions but also in localization or specificity, all of which may shape the mode of engagement with client proteins and subsequent chaperone activity. The canonical chaperones such as Hsp70, Hsp90, Hsp100, and others are ATP-fueled machines that assist folding of a wide range of cellular proteins and recognize differential conformational forms of the client proteins across their life span [4,27]. It was suggested that the expansion of proteomes from bacteria to mammalian cells led to an increase of more than six-fold in aggregation-prone proteins, requiring the evolution of chaperone and co-chaperone networks to facilitate a functional proteome in mammalian cells [28,29].

One of the cellular strategies for dealing with the expanded number of protein sequences and increased diversity in biochemical properties of evolved proteins is to increase the repertoire of chaperone-assisting proteins, i.e., co-chaperones. Co-chaperones are able to recognize misfolded proteins and ensure their interaction with the related chaperone via transfer to the substrate-binding domain of the ATP-dependent chaperone and enhancement of the chaperone’s ATPase activity [4,30,31]. Today, more than fifty different co-chaperones of human Hsp90 have been identified, varying in substrate specificity, catalytic activity, stress specificity, and even tissue specificity [32,33].

Another canonical chaperone, Hsp70, fulfills its function via assessment of a broad family of J-domain proteins (JDPs), including DnaJ in *E. coli*, Ydj1 [34] and Sis1 [35,36] in yeast, and DNAJB1/Hsp40 in mammalian cells [37,38,39]. It was shown that a crosstalk between different JDPs is essential for recognizing protein aggregates during proteotoxic stresses and aging [40]. One such J-protein is the mammalian ERdj5 (DNJ-27 in *C. elegans*), which comprises a typical JDP cysteine-rich domain fused with a thioredoxin-like domain [41]. ERdj5 is a functional reductase localized to the endoplasmic reticulum (ER), which is crucial for reducing undesirable disulfide bonds in misfolded proteins in the ER [42], tightening the redox and protein quality control functions together. Many of the known J-proteins harbor zinc (Zn)-binding domains that are crucial for their activity and might respond toward changes in cellular oxidation. In some of them, the redox status of the cysteines forming these Zn-binding regions defines anti-aggregation activity (e.g., ERdj3 in mammalian cells). Co-chaperones, including the J-proteins, have a crucial role in making the “life–death” decision and targeting the misfolded protein (instead of the folding one) to degradation by the proteasome [43,44].

To assist the ATP-dependent chaperones and their co-chaperones during stress conditions, other ATP-independent chaperones take on an essential role in maintaining a healthy proteome. This is particularly relevant under conditions associated with a drop in intracellular ATP reservoirs [45,46,47] (e.g., oxidative stress, mitochondrial dysfunction) or cellular locations depleted of ATP (e.g., bacterial periplasm). These are “holdases” or holding chaperones, which serve as the first line of defense in conditions leading to protein misfolding, for example Hsp33 [21], CnoX [48], and Get3 [49] during oxidative unfolding, HdeA/B [22,23] during acidic unfolding, and others. As chaperones may also target different aspects of protein quality control itself (e.g., protein folding, unfolding, assembly, disaggregation, etc.), they frequently work in tandem with different ATP-dependent chaperones to carry out their function. The ATP-independent activity and the ability to be a part of the cellular chaperone network serve as the foundation for the working cycles of redox-dependent chaperones, which maintain protein homeostasis during oxidative stress conditions.

Here, we will discuss the broad-scale relationship between cellular oxidation and proteostasis, focusing on the role of protein thiols as redox sensors and switches of the protein homeostasis network.

## 2. Cellular Oxidants: Origin, Targets and Benefits

The aerobic lifestyle has a proven advantage in efficient energy production. That said, it is also a major source of intracellular reactive oxygen and nitrogen species (ROS and RNS, respectively). From the quantitative aspect, oxygen-dependent ATP production in mitochondria is the largest contributor to the cellular ROS reservoir. As far back as the 1960s, Jensen and others showed that oxygen reduction in mitochondria leads to a flux of superoxide anions (O^•−^_2_) [50], which is further converted into hydrogen peroxide (H_2_O_2_) [51,52]. Formation of superoxide mainly occurs in complexes I and III in the respiration electron transfer chain (ETC) as a byproduct of oxygen reduction [53,54]. Since superoxide is a very reactive yet unstable radical, it is rapidly converted into hydrogen peroxide spontaneously or enzymatically through distant superoxide dismutase (SOD) enzymes [55,56,57].

In addition to mitochondria, cellular ROS is produced by a variety of enzymatic reactions in a regulatory way. This includes oxidative protein folding in the ER [58], lipid oxidation in peroxisomes [58], the inflammation process via activity of a diverse family of nicotinamide adenine dinucleotide phosphate (NADPH) oxidases (NOX) [59,60,61], oxidation of monoamine neurotransmitters using monoamino oxidases (MAO) [62], and many other processes [63,64].

Another group of chemically reactive endogenous molecules is that of reactive nitrogen species (RNS), which are mainly derived from nitric oxide (NO) interactions. One of the major sources of cellular nitric oxide molecules is the enzymatic activity of a family of nitric oxide synthetase (NOS) enzymes involved in cell signaling and immune defense [65,66]. Reactions between NO, ROS, and metals result in derivatives of other reactive molecules such as peroxynitrite (ONOO-), NO radicals (NO-), S-nitrosothiols (SNOs), and others [67,68]. The origins and crosstalk between RNS and ROS are well detailed in the following reviews [69,70].

Not surprisingly, organisms have found a way to use ROS and RNS reservoirs for cellular activity, maintaining a delicate balance between producing the needed oxidants at the right time and place, while detoxifying others. Impairment of this balance leads to oxidative and nitrosative stress, causing potential damage to macromolecules in cells.

One of the first pieces of evidence for the positive role of ROS was made by Babior, Kipnes, and Curnutte in 1973, by demonstrating that activated phagocytes (leukocytes and granulocytes) produce superoxide during phagocytosis [71]. This suggested that ROS can serve as a native antibacterial agent during the immune response. Later, this phenomenon led to the uncovering of a fundamental class of protein complexes (NOXs) that actively produce superoxide molecules to kill pathogens. Since then, other ROS-dependent pathways have been discovered and characterized, many of which utilize hydrogen peroxide as a molecular messenger or mediator of signaling pathways, including neurotransmission [72], cell proliferation and inflammation [57,73], and protein quality control, which will be discussed in further detail below.

Like ROS, RNS are important chemical messengers. In 1992, Daniel Koshland named nitric oxide as “the molecule of the year” [74] due to its important role in medicine (i.e., maintaining blood pressure), the immune response, and neuron function. Six years laterin 1998, Ferid Murad received the Nobel Prize for the discovery of the beneficial role of NO in cell transduction through activation of soluble guanylyl cyclase (sGC), which produces the signaling molecule 3′,5′-cyclic guanosine monophosphate (cGMP). Numerous modern studies have subsequently defined the molecular basis of NO and other RNS in cellular metabolism, apoptosis, and cellular proliferation through post-translational site-specific S-nitrosylation of signaling proteins. The detailed molecular basis of RNS and ROS-mediated cell signaling processes are well reviewed in [69,75,76,77].

Thus, cells have developed an elegant way to recycle ROS and RNS byproducts and convert them into biological readouts through site-specific oxidation of the target proteins, especially in cysteine thiols.

## 3. Cysteine Thiols: The Central Components of Redox-Regulation of Proteostasis

The cellular redox status sits at the junction between protein homeostasis and the global stress response, in large part through the coordinated role of molecular chaperones and the degradation machinery in maintaining protein homeostasis [4]. More specifically, oxidative stress has long been established as one of the primary sources of different forms of cellular damage, whether through direct modifications on individual cysteine residues or perturbations of existing protein complexes through disulfide-bridge formation, which all might result in protein misfolding and aggregation [78,79]. Numerous age-related diseases and disorders have been associated with changes in both redox and protein homeostasis, bridging these two defense systems together [80,81]. This includes neurodegenerative disorders such as Alzheimer’s and Parkinson’s diseases, which are associated with the buildup of cellular oxidants, mitochondrial dysfunction, and accumulation of “toxic” misfolded proteins (TAU and α/β amyloids in Alzheimer’s and α-synuclein in Parkinson’s diseases) into cellular inclusion bodies, followed by progressive death of specific neural cells [82,83,84,85,86,87].

This comes alongside the beneficial role of oxidants mediating “normal” signaling and biological responses triggered by cellular oxidation [88,89], which raise the question as to how cells maintain the balance between oxidation and proteostasis at large.

One of the major and most studied mechanisms for sensing and protecting cells against oxidation is utilizing the reactivity of cysteines, specifically sulfhydryl (thiol) groups, serving as oxidant sensors. Cysteines are highly reactive for oxidation, and this sensitivity makes them “sweet and sour” spots in cellular proteins. On the one hand, the accumulation of intracellular oxidants results in undesirable thiol (over)oxidation, leading to the addition of negatively charged modifications to the protein in the form of sulfenic (RSOH) or sulfinic acids (RSO_2_H). It may also introduce non-native disulfide bonds affecting protein structure and stability or even a covalent crosslink with other macromolecules. On the other hand, while the majority of proteins might lose function upon non-specific oxidation, other types of proteins use a site-specific thiol oxidation as an “on–off” switch for rapid function regulation. This mode of activation is similar to other post-translational modifications, which immediately change the chemical properties of a protein, leading to a loss or gain of function. In recent years, many such “redox switches” (or “thiol switches”) [90] have been identified as redox-regulated proteins, spanning from bacteria to mammals and fulfilling diverse biological functions [91,92,93].

Reactive thiols of redox-regulated proteins usually have distinctive chemical properties that define reactivity towards oxidants and reversibility. They can be modified in various ways: sulfenylation, nitrosylation, glutathionation, persulfidation, and disulfide formation, responding to different oxidants (Figure 1). Some modifications may emerge as either reversible (e.g., palmitoylation, sulfenic acid) or irreversible (e.g., prenylation, sulfinic, and sulfonic acid).

There are also specific cysteine-containing motifs, which may be involved in redox switch activity, such as derivatives of the CxxC sequence motifs (where “x” represents any amino acid), metal-binding sites, or cysteines that are available for disulfide bridge formation. Among the first identified and very well-characterized redox switches are antioxidant proteins, which use rapid changes in the redox state of their catalytic cysteines to restore the redox status of cellular proteins (e.g., thioredoxins, glutaredoxins) or detoxify an excess of ROS and their byproducts (e.g., peroxiredoxins) [94,95]. These proteins use catalytic cysteine residues in highly specific and regulated manners, often through conserved motifs that determine cysteine thiol-based interactions. It was shown that sensitivity of thiol cysteines in these canonical antioxidant proteins is defined by their position in the protein sequence and protein environment, which can affect their pK_a_ value and redox potential. Location near basic residues decreased the pK_a_ of the redox-sensitive thiol, lowering the thiol pK_a_ value from 9 to 5–7, turning it into an efficient nucleophile [96]. The canonical antioxidant proteins utilize reversible oxidation–reduction cycles of their highly conserved pair of redox-sensitive cysteines to restore the redox states of cellular proteins and mediate redox-dependent signaling events in cells. The mechanisms and roles of thiol modifications in cell survival and apoptosis were reviewed in detail in Benhar (2020) [97]. Recent studies have also shown how conformationally adjacent regions may impact the glutaredoxin activity, beyond the cysteine residues themselves [98]. Liedgens et al. determined that various additional residues (both motif-adjacent and conformationally available) are part of the glutathione-scaffold site, mutation of which alters either the reductive or oxidative half reaction, pointing to an expanded involvement of nearby regions in functional cysteine oxidation.

Interestingly, despite a large number of canonical antioxidant enzymes sharing similar functions and a wide range of thiol modification types, these enzymes are not redundant in their substrate selection, taking care of a defined set of client proteins. Moreover, proteomic profiles of thiol modifications show that they are site- and type-specific, with minimal crosstalk between them [99,100,101,102]. In contrast, the proteostasis network relies on a redundant function of its members, ensuring proper folding and degradation of similarly misfolded proteins [4,103].

The thiol-redox switch group may be present in proteins of varying roles, however there are notable recent advancements in identifying proteostasis-related redox switches, opening up a new era of understanding molecular mechanisms of redox regulation of the proteostasis function in cells [13,104] (Figure 2). Discovery of a dual function of peroxiredoxin proteins was one of the major breakthroughs in understanding the link between anti-aggregation and antioxidant activities. Peroxiredoxins are antioxidant enzymes that reduce cellular peroxide or peroxynitrite derivatives into harmless water molecules [105]. However, severe oxidative stress converts some of these enzymes (e.g., Tsa1 in yeast, Prx1, and Prx2 in humans) into powerful ATP-independent chaperones without the antioxidant activity [106,107]. This transition is triggered by overoxidation of one of the catalytic cysteines, which forms sulfinic acid coupled with substantial oligomerization changes. Along with conformational changes, thiol-specific oxidation might recruit other members of the proteostatic family in order to facilitate refolding of the misfolded proteins captured by peroxiredoxin. For example, hyperoxidation of its Cys48 in yeast peroxiredoxin Tsa1 recruits Hsp70 chaperones and the Hsp104 disaggregates to H_2_O_2_-induced aggregates [108]. However, the overoxidation of peroxiredoxins can be restored by sulfiredoxin enzymes, which reduce cysteine–sulfinic acid in an ATP-dependent manner [109]. This restores the antioxidant activity of peroxiredoxins during non-stress conditions [110,111]. It is important to note that the chaperone activity of peroxiredoxins is not limited to oxidation and can occur during other protein unfolding stresses, such as heat shock [112,113], metal deficiency [107], and acidic stress [114]. Additional recent advancements showcase the importance of persulfidation as a conserved cysteine modification that plays a role in protecting cysteine residues from irreversible overoxidation during oxidative stress [115]. This was specifically shown through modifications of mouse peroxiredoxins as mediated by thioredoxin-related proteins (TrxR1 and TRP14), with an additional protective effect on several redox-sensitive proteins with roles relating to protein quality control pathways, some of which will be discussed in further depth (e.g., PTP1B [116], HSP90 [4], and KEAP1 [117]) [118]. Taken together, these place peroxiredoxins at the center of both protective and potentially harmful roles (e.g., as treatment against ionizing radiation and upregulation in cancer cells, respectively) [119,120].

Meanwhile, there remain many other redox-sensitive proteins without explicit antioxidant behavior, which actively interact with the oxidative stress response in some form and play a direct role in proteostasis (Table 1). These include a wide range of chaperones that are necessary for assisting proteins that become damaged as a result of oxidative stress, whether through direct refolding (partial or complete), “holding” functions that may interact with the misfolded proteins along with other chaperones, or indeed relaying misfolded proteins to degradation itself. In this way, members of the protein quality control (PQC) system may themselves be recruited during oxidative stress conditions in order to maintain the necessary homeostasis. This regulation is critical, as a breakdown of the PQC system can lead to the formation of misfolded proteins as well as the potential toxic protein aggregates that may follow [5,121].

Subsequent research of thiol-redox switches in a proteostasis context revealed an important property of some of the thiol-redox switches—a requirement to undergo substantial conformational rearrangements regulated by its redox status in order to gain anti-aggregation activity [104]. One such redox-regulated chaperone, which uses protein plasticity for its activation, is a highly conserved ATP-independent holdase chaperone, Hsp33. Hsp33 is a predominantly bacterial chaperone, also found in unicellular pathogens such as *Trypanosoma* and *Leishmania* [122], which protects microbes against a wide range of oxidants similar to those applied by the host immune system [21]. Exposure to oxidants or chlorine species (e.g., HOCl) triggers large conformational changes and exposure of hydrophobic regions involved in the anti-aggregation activity [91,123,126,127]. Upon return to reducing conditions, Hsp33 refolds due to tight and highly specific interactions between its domains [91], leading to a destabilization of the bound client protein and transfer to the foldase chaperone system, DnaK/J [126]. The reduction in Hsp33 is mediated by physiological antioxidant enzymes, such as thioredoxins and glutaredoxins. While constitutive (non-redox dependent) activation of Hsp33 can be easily achieved by either chemical denaturants, extreme heat, or single mutations [128], its inactivation mechanism is highly conserved and difficult to alter [91]. Thus, the evolutionary path of Hsp33 appears invested in providing unique structural features enabling reversibility of its activity, which prevents constitutive binding with misfolded proteins during normal, non-oxidative conditions.

Another example of a redox-sensitive chaperone is the recently identified CnoX bacterial chaperone, which protects cells against hypochlorous acid (HOCl) [48,129]. CnoX is reversibly activated by site specific chlorination, which leads to exposure of hydrophobic regions, most probably crucial for anti-aggregation activity. Similar to Hsp33, CnoX transfers its misfolded client proteins to the ATP-dependent system, GroEL and DnaK/J. It is still unknown if conformational changes on CnoX are coupled with this process. Interestingly, as with peroxiredoxins, CnoX from specific bacterial strains (including *C. crescentus)* acts as an oxidoreductase, utilizing its CxxC domain located in a thioredoxin fold. It was shown that *C. crescentus* CnoX is a constitutive chaperone with oxidoreductase activity, raising an intriguing evolutionary question as to the reason for this loss of function in specific bacterial strains such as *E. coli*.

Furthermore, the yeast Get3 (TRC40 or Asna1 in mammals) is an additional “moonlighting” redox switch with multiple functions [49]. Under reducing conditions, Get3 serves as a delivery protein together with other Get proteins, ensuring post-translational delivery of the thiol-anchoring (TA) proteins to the ER in an ATP-dependent manner. Under oxidative conditions, similar to Hsp33, the oxidation of Get3 leads to the formation of disulfide bonds and a structural rearrangement followed by oligomerization, converting the oxidized form of Get3 into a general, ATP-independent chaperone. These three examples demonstrate some of the diversity among redox-sensitive chaperones and the ways in which these proteins intersect with additional protein quality control pathways.

## 4. Integrative Approaches for Discovering New Redox Switches in PQC

Intrigued by the moonlighting function of redox-regulated proteins, which apply structural plasticity for their redox-sensitive function, Erdos et al. developed a computationally based structural prediction web server IUPred2A that points to cysteine-containing regions, which might undergo disorder-to-order transitions in response to changes in their redox status [130]. IUPred2A is a new generation of a very well-established prediction algorithm of intrinsically disordered proteins (IUPred), which was trained on outcomes of redox proteomic studies and sets of known redox-switch proteins to obtain valid predictions of potential redox switches. This study estimated that around 5% of proteomes harbor redox-sensitive thermobile regions, while viruses are among the most divergent proteomes regarding the predicted number of redox-regulating proteins. Moreover, this study suggests that metal binding regions are the most frequent among conditionally disordered regions including many experimentally validated redox-sensitive DnaJ co-chaperones such as Ydj1 and others [130].

To go beyond the individual cases and bioinformatic prediction, the advances in thiol labeling coupled with quantitative proteomics enable identification and accurate estimation of redox-sensitive thiols in diverse protein families. These mass spectrometry-based methods utilize differentially labeling thiol-specific alkylating reagents (e.g., isotope-coded affinity tags (ICAT), cysteine-reactive phosphate tags, tandem mass tags (TMT), and dimedone-based probes, as well as relatively simple N-ethylmaleimide (NEM) and iodoacetamide (IAM) reagents [131,132,133,134,135,136,137]), which are used to broadly map out cysteine oxidation in a quantitative manner. These wide-scale assessments may then be applied to specific cellular functions or conditions (e.g., different stresses), though these do not necessarily address proteostasis directly. However, these varying methods could easily be applied to study the redoxomes (i.e., redox proteomes) of individual systems within the PQC under diverse conditions. Undoubtedly, identification of the specific redox-sensitive thiols is only the first stage, to be followed by extensive biochemical analysis and validation in the context of physiologically relevant conditions.

Moreover, modern high-sensitivity mass spectrometry, genomic sequencing, and RNA-sequencing methods have enabled identification of proteins more generally associated with changes in global oxidation [13,138,139,140,141]. While this does not guarantee identification of individual thiol switches, it serves as a useful guidepost for defining proteins involved in maintaining redox homeostasis in some form. Indeed, a comparison of the proteomes of endogenously oxidized versus reduced cells revealed that reduced cells had relatively higher levels of proteins involved in folding pathways and the proteasome, as well as proteins involved in stress granule formation [140]. All in all, large-scale quantitative methodologies seem poised to fill in many of the gaps in our current understanding of redox switch regulation of protein homeostasis.

## 5. Cysteine-Mediated Modifications: An Efficient Mechanism to Regulate Signal Transduction and Protein Localization in Cells

Direct oxidation is not the only form of modification found on cysteine residues, nor is it the only form of cysteine-mediated regulation of proteostasis. Cysteines are primed to undergo an extraordinarily wide range of modifications, from disulfide bonds to S-nitrosylation through to reversible fatty acid modifications such as palmitoylation, many of which have been extensively studied through mass spectrometry-based proteomic techniques (Table 2) [142]. Due to the importance of post-translational modifications in proper protein folding, function, and secretion, cysteine modifications are crucial for different cellular regulatory processes and healthy cellular growth. Despite the difficulty in studying lipid modifications on a large scale, individual case studies have been able to identify specific cysteine residues involved in protein quality control on varying levels.

Among these different potential modifications, palmitoylation is a major cysteine-specific modification responsible for regulating signaling pathways within the cell. Here, either the presence or absence of palmitoylation may play a role in protein localization or in determining protein export from the ER [148]. This has been found in many different forms and utilizing different mechanisms. In some cases, palmitoylation specifically enables protein export from the ER (e.g., Wnt signaling protein LRP6) [149]. However, palmitoylation may also regulate protein aggregation within the ER prior to export, as seen for proteins such as yeast chitin synthase, Chs3 [145] (Figure 3), or alternatively, stabilize proteins for aggregation as in human adult-onset neuronal ceroid lipofuscinosis-causing CSPα mutants [150]. Whether playing a role in general protein export or through involvement in direct folding or aggregation-prevention, the presence of the cysteine-based palmitoylation is, thus, critical in determining proper protein function, localization, and behavior.

Such modification-based regulation of protein transport is not limited to the ER and has also been identified in the Golgi. Known palmitoylated cargo proteins were found to have a higher rate of transport within the Golgi as compared with un-palmitoylated mutant proteins [151]. Taken alongside an increasing awareness of Golgi quality control (GQC) as an independent and multipronged system for quality control within the secretory pathway [152], the potential for palmitoylation (or other cysteine-based modifications) as a PQC-regulating modification remains intriguing, albeit under-explored.

This similarly applies to other cysteine-specific modifications such as prenylation, which plays a central role in cell cycle and cancer regulation [153]. Once again, cysteine modifications have been found to play an important role in the quality control. For example, the cancer-associated chaperone SmgGDS has been found to associate differently with prenylated RhoA (Figure 3), and that different splice variants are in fact structurally designed to accommodate cysteine modification [147]. This suggests that prenylation (including farnesylation and geranylgeranylation specifically, as the lipid modifications) may be viewed as both structure- and function-modifying. In another example, altering a cysteine-based lipid modification in the *Schizosaccharomyces pombe* yeast Rho1 was recently shown to trigger the formation of protein aggregate centers (PACs) under mild heat stress [154]. By blocking access to the prenylation target, Rho1 was found to no longer localize to the plasma membrane at permissive temperatures and the aggregate-like PACs at 37 °C. These PACs, in turn, demonstrated a widespread rearrangement of PQC members within the cell, including the presence of Hsp70- and Hsp40-family chaperones, Hsp104, and more. This demonstrates, in part, the importance these modifications play in the PQC system specifically through regulation of proper protein folding and localization, as well as protein–protein interactions more generally. Given the widespread nature of the different lipid modifications in diverse eukaryotic proteomes, we may speculate that many other, yet undiscovered proteins similarly require cysteine-specific lipid modifications in order to fully mature and function under healthy conditions.

Other types of cysteine modifications may play a regulatory role in protein quality control as well. S-nitrosylation is well studied in the context of RNS, as previously described in detailed reviews on its implications for both cardiovascular and neurodegenerative diseases [155,156]. The latter is particularly interesting as a model for S-nitrosylation involvement in protein folding and aggregation. S-nitrosylation also occurs on chaperones such as protein disulfide isomerase (PDI) [157,158] and the mitochondrial TRAP1 [143] at the center of quality control mechanisms, emphasizing the varied relationship S-nitrosylation may have with protein quality control mechanisms (Figure 3). Interestingly, both in vitro and in vivo PDI retained fairly stable levels of S-nitrosylation, with only slow-reducing reversibility by glutathione, which may suggest a more direct involvement of S-nitrosylated PDI in neurodegenerative disorders [144,157].

In another example, studies in plants have also identified S-nitrosylation of several conserved cysteine residues in the *Nicotiana tabacum* Cdc48 [159]. Among these, Cys526 was found to be involved in ATPase activity and protein conformation (without affecting protein structure at large), alongside redox regulation of the conserved cysteine in the mammalian homolog VCP [160]. Interestingly, another identified S-nitrosylated conserved cysteine residue in plant Cdc48 was found to undergo palmitoylation in the human homolog [161]. Together, these point to the ways in which S-nitrosylation specifically and cysteine modifications more generally regulate proteostasis on individual protein levels. New methodologies combined with an increased proteomic sensitivity introduce the potential for large-scale screens to identify additional cysteine-specific modifications at varying stages of proteostasis.

## 6. Thiol Editing in the ER Is Mediated by Molecular Redox Switches

Cysteine-specific modifications that define protein activity or stability are not alone in bridging proteostasis and redox regulation. The ER itself plays a pivotal role in the PQC system, as this is frequently the site of protein folding and assembly within the cell [121,162]. Importantly, the ER is also where many proteins undergo redox modifications, which themselves often tie into PQC more broadly. Disulfide bond formation in the ER is largely mediated by Ero1 (ER oxidoreductin) and protein disulfide isomerase (PDI), as well as additional proteins belonging to the PDI family [163,164]. As previously mentioned, oxidative modification of cysteine residues in PDI itself has been linked to ER stress, for example in human neurodegenerative diseases [144,165].

This balance between redox homeostasis and the ER is further played out through the unfolded protein response (UPR) (Figure 2). The UPR follows the failure of the ER-associated degradation (ERAD) pathway to clear toxic or aberrant proteins [166], leading to a cascade of different signaling pathways, which may ultimately result in apoptosis [3]. Different proteins within UPR signaling have been found to contain redox-sensitive regulation; indeed, the UPR at large has been suggested to be broadly regulated by changes in oxidation [167] and during aging [168]. Among its major players, the ER stress-activated ATF6 transcription factor has been suggested to undergo a redox-dependent regulation in its response to an increase in misfolded proteins within the ER [169]. Ire1, another major UPR factor that is present in mammals, worms, and yeast, has also been found to contain an evolutionarily conserved cysteine (Cys663 in *C. elegans*) residue, which regulates its activity [170,171], while a different cysteine residue (Cys148) has been implicated in a potential cysteine oxidation-dependent interaction with a member of the PDI family in *C. elegans* [172]. The roles these prominent UPR factors play again demonstrate the intimate relationship between proteostasis quality control systems and individual cysteine residues.

## 7. Regulation of Protein Degradation during Oxidative Stress

The relationship between cysteine–thiol regulation within the ER and ER-related degradation pathways demonstrates the important role cysteines may play in mediating protein “preparation” or identification for degradation. Damage to proteins following oxidative stress draws a clear connection between protein oxidation and proteostasis, largely due to the importance of clearing these aberrant proteins from within the cell. This process may ultimately lead to degradation pathways, when no refolding alternative is available [13,17]. Various studies have identified redox-based regulation of different subunits or cofactors of the proteasome, including through specific cysteine modifications [173,174]. Moreover, the proteasome itself has demonstrated a degree of oxidative sensitivity, with differences between the 20S and 26S complexes in terms of maintaining activity during oxidative stress across yeast and mammalian models [175]. That the proteasome has been found to directly interact and degrade certain oxidized proteins, thus, only strengthens the regulatory relationship between the oxidative stress response and proteostasis at large [176].

Proteasome-mediated degradation of irreparably misfolded proteins is, therefore, one of the most important and well-regulated stages of the oxidative stress response [177]. The proteasome is responsible for the majority of ubiquitin-dependent and -independent degradation of oxidized proteins, through either the 26S or 20S proteasome. Under regular conditions, a vast portion of cellular proteins that are targeted for proteolysis are ubiquitylated and degraded by the 26S proteasome (Figure 4). The 26S proteasome—a 2.5 MDa complex—comprises a 20S core segment, with two 19S regulatory particle (RP) ATP-dependent segments wrapping the 20S from both sides, which can asymmetrically attach to the 20S core particle [178]. More regulatory particles are attached to the 20S proteasome, including the proteasome activator PA200 in mammals (Bml10 in yeast) and the PA28/11S particle [179].

Ubiquitylation of proteins by itself is insufficient for protein degradation by the proteasome, requiring the presence of unfolded and conformationally available domains in the substrates. Two highly important steps determine substrate fate: (1) a preliminary reversible step of ubiquitylation, which is dependent on ATP binding, and (2) recognition of unfolded regions in the client protein. The second is an engagement step for a cascade of substrate processing in the proteasome, leading to the final degradation of the protein [180].

The 19S particle is the recognition subunit of the 26S proteasome and is composed of several subunits itself, assembling the base and the lid (Figure 4). Rpt1-6 (regulatory particle triple-A protein) is responsible for ATPase activity, alongside non-ATPase subunits (Rpn) 1, 2, 10, and 13, forming the base. The lid, meanwhile, is composed of Rpn3, 5-9, 11, 12, and Sem1 [181]. Rpn10 and 13 both bind ubiquitin chains and ubiquitin-like (UBL) domains alike [182,183]. Although these two subunits are the main recognition particles of ubiquitinylated proteins or UBLs, the 19S particle has additional subunits capable of initially binding to substrates [183]. Shuttle proteins are crucial just prior to the initial binding to the proteasome, delivering targeted proteins to the proteasome. One such “shuttle” protein is the previously mentioned Cdc48/p97/VCP ATPase, which is a key factor in collecting and guiding targeted proteins to the proteasome [184]. Another important shuttle protein is UBQLN2, which has been found to have amyotrophic lateral sclerosis (ALS)-causing mutations [185,186].

Under normal conditions, the different components of the proteasome undergo several conformational changes. After its initial recognition and binding to unfolded regions, the 19S faces a widening of its ATPase pore and correspondingly of the 20S as well. Three consecutive ATP-dependent processes subsequently occur: substrate unfolding, gate opening in the 20S, and protein translocation [187]. The 20S core particle is itself made up of two inner β-rings (comprising seven β-subunits) that operate the proteolytic activity, while two outer α rings (comprising seven α-subunits) tightly control the gate opening for substrate entry, which prevent unwanted proteins from being degraded. Meanwhile, β1, β2, and β5 perform caspase-like, trypsin-like, and chymotrypsin-like activity, respectively [188].

Under oxidative stress, however, the two 19S caps dissociate from the main 20S core segment, leaving behind only the 20S core [189] (Figure 4). This process is mediated by the Ecm29 protein in yeast (encoded in mammalian cells by the KIAA0368 gene), as shown by crosslinking coupled with mass spectrometry. Specifically, mutants of Ecm29 displayed reduced dissociation of the 26S proteasome and subsequent sensitivity to H_2_O_2_ stress, emphasizing the importance of this proteasome dissociation within the 26S complex triggered by oxidative stress [189,190,191].

Moreover, the previously discussed canonical ATP-dependent chaperone Hsp70 is also involved in the 26S disassociation upon oxidative stress. Not only does it help in complex decoupling, it withholds the 19S cap and reconstitutes the complex after the return to non-stress conditions [192].

The subsequent degradation of proteins by the 20S is coordinated by NAD(P)H:quinone-oxidoreductase-1 (NQO1) and protein deglycase DJ-1, which are similar in structure, both sharing a Rossman fold and activity regulated by oxidative stress [173]. NQO1 and DJ-1 bind directly the 20S proteasome and enhance protein degradation specifically during oxidative stress conditions. The activity of these factors has feedback loop regulation: DJ-1 is involved in the Nrf2-dependent oxidative stress response that leads to upregulation of NQO1, an apo form of which is degraded by the 20S proteasome [193,194].

An additional recent study using a bioinformatic analysis utilized the structural and sequence similarity of DJ-1 and Nrf2 to extend the family of 20S regulators (named catalytic core regulators or CCRs), adding 17 other potential regulators into this club including some which have been experimentally verified [195]. Specifically, several of the identified CCRs were found to affect 20S proteasome activity in vitro, while their overexpression in vivo also demonstrated 20S proteasome inhibition.

Abi-Habib et al. showed that the dissociation of the 19S gives the green light for a direct interaction of oxidized and unfolded proteins with the 20S particle. This occurs through exposure of hydrophobic regions in an ATP-independent manner, which is especially central as oxidative stress leads to ATP depletion [196]. With that being said, depletion of ATP is itself directly connected to the disassembly of the 26S proteasome complex. The stability of the complex can be maintained by the NADH molecule as a compensating agent. Five subunits among those of the 19S complex are suspected of having an NADH binding motif, clarifying the important role NADH has under depletion of ATP conditions, as under oxidative stress [197].

Another subtype of proteasomes found in cells is the immunoproteasome (20S IP) (Figure 4). The 20S IP has a similar structure to the 20S proteasome, but instead of the β1,2 and 5 subunits, it comprises β1i,2i and 5i subunits, expression of which are induced by IFN-gamma in mammals [179]. The β5i plays a critical role in the degradation of oxidized proteins alongside intrinsically disordered proteins [196]. Additionally, an alternative ATP-independent 11S cap or PA28 multimer is expressed under oxidative stress. Overexpression of PA28 showed increased protection against oxidative stress-induced apoptosis and clearance of oxidized and misfolded proteins by the proteasome [198]. This was done using a combination of DNA fragmentation and degradation assays to assess degradation of GFPu, a known proteasome substrate [198].

Many different oxidative stress-induced post-translational modifications have been identified on the proteasome, including carbonylation, glycoxidation, lipoxidation, and glutathionylation, affecting the stability of the proteasome assembly [199]. S-glutathionylation of cysteines within the α5 subunit was also shown to be involved in redox regulation of the proteasome, increasing proteasomal activity by supporting ring opening of the α subunits [174].

These findings highlight the degree to which different aspects of proteasomal degradation rely on cysteine oxidation as a means of functional regulation. In addition to its role in degrading oxidation-damaged proteins, the proteasome itself undergoes significant rearrangement during oxidative conditions, which shape its interactions and mechanisms of behavior. This proteasomal redox sensitivity ultimately appears to be an integral part of the global redox homeostasis, and it seems likely that future research will clarify this role even further.

## 8. Protein Degradation by Redox Sensitive Proteins

While the relationship between protein degradation and the cellular redox state is frequently thought of through the lens of oxidation-associated damage, increasing studies point to direct redox regulation of ubiquitin and ubiquitin-like degradation. This begins with the relationship between oxidation and E1/E2 ubiquitin enzymes. Initial binding of the ubiquitin molecule to the E1 ubiquitin-activating enzyme is specifically triggered through formation of a thiol ester bond [200]. Subsequent transfer of the activated ubiquitin to an E2 ubiquitin-conjugating enzyme is also mediated by an active cysteine residue within the E2 [201].

Furthermore, numerous ubiquitin E3 ligases have been found to contain reactive cysteines that regulate ubiquitylation. Some E3 ligases or complex members undergo cysteine modifications, which in turn, affect ligase activity or interactions. A well-studied example of a directly redox-regulated E3 ligase adaptor is Keap1, which has multiple reactive cysteine residues determining interaction with the redox-associated transcription factor Nrf2, including in response to varying degrees of mild H_2_O_2_ treatment (100–400 μM) alongside other inducing conditions [117,202]. Thus, cysteine oxidation not only controls ubiquitylation and subsequent proteasomal degradation of a protein target, it can also serve as a regulation mechanism for ROS sensing [203].

A well-studied E3 ligase in the context of different redox regulation or cysteine modification is Parkin [204,205,206]. Parkin belongs to a family of RING-between-RING (RBR) ubiquitin ligase enzymes, further containing an in-between-RING (IBR) domain [207]. These domains are considered important for ubiquitylation of Parkin substrates and interaction with ubiquitin-conjugating enzymes [208,209,210]. RING0, RING1, RING2, and the IBR domains in the Parkin protein are cysteine-rich subdomains, which bind up to eight Zn^2+^ ions [211]. Moreover, Parkin is translocated to the mitochondria under oxidative stress in the absence of DJ-1 [212]. As previously discussed, DJ-1 is a known cellular regulator of ROS [213] and its deficiency leads to the appearance of fragmented mitochondria. This phenotype is rescued by Parkin or PTEN-induced kinase 1 (PINK1) expression [214].

However, ubiquitin enzymes or ligases are not the only redox-sensitive proteins when looking at ubiquitin-related pathways. Interestingly, similar redox-regulation has been identified for both SUMO-associated subunits as well as NEDD8/Rub1 [215]. For SUMOylation, oxidation of specific cysteines between the E1 and E2 enzymes Uba2 and Ubc9 have been directly implicated in the formation of a disulfide bridge between the two enzymes, which in turn, inactivates them [216]. This was later found to play a direct role in the cellular redox response, such that variants that directly affected the disulfide bond’s stability severely altered cell survival during oxidative stress [217].

Redox regulation also plays an important role in the interplay between SUMOylation and ubiquitylation itself. The mammalian SUMO protease SENP3 is constantly regulated by the ubiquitin-proteasome system, with a cysteine modification controlling its association with the chaperone Hsp90 and, thus, its degradation [218,219]. Additional reactive cysteines have been identified in other SUMO proteases, leaving open the possibility of a broader role for cysteine oxidation in SUMOylation across different organisms [220,221]. A similar redox-regulatory role also exists for another ubiquitin-like pathway, NEDDylation/rubylation, with changes in rubylation following oxidation of the yeast cullin Cul1 under both exogenous (4.4 mM H_2_O_2_) and endogenous oxidative conditions [222].

Taken together, these results place cysteine-mediated redox regulation of ubiquitin/ubiquitin-like degradation pathways at the center of varying mechanisms and across different model organisms.

## 9. Aging, Subcellular Localization, and Cysteine Oxidation

Aging is another process that raises particularly interesting questions as to the relationship between protein homeostasis and thiol oxidation. The link between proteostasis and aging has been well established [15,223], with much research into the relationship between proteostatic stress and different hallmarks or types of aging (e.g., replicative aging in yeast [224], diseases in various organisms [8,223]). This is of particular interest when studying neurodegenerative diseases, which are predominantly found in older patients and have been shown in correlation with a breakdown of the proteostasis network [14,223]. This further correlates with the well-studied changes in cellular oxidation during aging across different models, with correlations between replicative aging in yeast and changes in cellular oxidation as well as during chronological aging [140,225,226], though there are additional organisms, which suggest different mechanisms. For this reason, studying cysteine oxidation during aging may provide a relevant context for understanding the pathologies of highly prevalent aging-related disorders.

Several cysteine oxidation mapping methodologies (predominantly proteomic) have been applied in the context of cellular or system-wide aging. With the understanding that the PQC undergoes significant changes during aging, these studies may demonstrate the tight relationship between PQC and redox switches specifically, with many proteins and pathways identified as having aging-dependent changes in cysteine oxidation. Moreover, many of the observed changes are in proteins that are directly involved in regulating protein homeostasis at large, and as such suggest that these thiol switches serve functional redox-dependent roles within the cell, possibly in PQC itself [132,227].

Different studies focus on cysteine oxidation during aging from multiple perspectives. In yeast, for example, a varied group of proteins was identified based on cysteine residues, which underwent “early” oxidation, prior to the global redox collapse associated with chronological aging (under both normal and caloric restriction conditions) [227]. This experiment was conducted using redox proteomic methodologies to track individual thiol oxidation during extended stationary growth, as compared to “full” thiol oxidation under 500 μM H_2_O_2_ treatment [102]. Of the early oxidized cysteines, several were found in chaperones or PQC-related proteins (e.g., Cdc48 [228,229,230], Ydj1 [231,232], Ubc4 [233]), with others belonging to the ribosomal 40S and 60S complexes. This points toward potential functional roles for these cysteines in regulating proteostasis.

Intriguingly, the early oxidized cysteine residue in Cdc48 is the same conserved cysteine identified as undergoing different modifications in other organisms, as previously discussed. This comes alongside additional research, which has found that the highly prevalent, ERAD-associated AAA-ATPase has a functionally redox-sensitive cysteine in one of its ATPase domains, with ATPase activity specifically regulated by oxidative stress [160]. Furthermore, Ydj1 and additional ribosomal 40S and 60S subunits are predicted to have conditionally unfolded regions [130], which may further point to functional changes following potential cysteine oxidation. Ydj1′s cysteines have more broadly been implicated in H_2_O_2_-induced oxidation and subsequent inactivation [102,108], which leaves open intriguing questions regarding a potential role in regulating proteostasis at large. While these do not directly address aging, they remain curious avenues for future research and raise questions as to the roles individual cysteine residues may play in regulating the redox response under different growth conditions.

However, further additional studies into the relationship between cysteine oxidation and aging suggest the involvement of several other pathways and individual proteins in maintaining protein homeostasis. A wide-scale proteomic screen of reversible tissue-specific cysteine oxidation in mice (establishing the Oximouse technique) demonstrated the presence of cysteine oxidation networks with stark changes in thiol oxidation during aging, with notable differences mapped to human disease-associated proteins [132]. Many of the changes were tissue-specific and were interestingly not limited to oxidation alone, with cases of thiol reduction during aging as well. On a mechanistic level, tissue-specific cysteine oxidation has been found to play a functional role in the stabilization of the human lipid biosynthesis regulator Insig-2 [234]. These pathways resemble those regulated by other proteins with potential redox switches, such as the Cdc48 ergosterol degradation pathway [235] and Txnip-regulated lipid homeostasis [236]. Other identified redox networks also included tRNA multi-synthetase complex members, which suggested oxidation-based regulation of translation. Meanwhile, the presence of thiol reduction during aging (alternatively viewed as the “loss” of a highly oxidized site in older mice) raises several new questions in terms of oxidative changes during aging on the whole, which will likely need to be studied to a greater degree in the future.

Oxidation of cysteine residues across unique pathways is not limited to changes during aging. Studies have been able to identify large-scale thiol oxidation in yeast, finding that proteins with highly oxidized cysteine residues were preferentially localized to organelles such as the mitochondria, ER, and vacuole [92]. Furthermore, oxidative treatment with 1 mM H_2_O_2_ in the same study revealed specific thiol oxidation in proteins linked to translation, ribosome biogenesis, and subsequently, ribosomal degradation mechanisms [92]. Taken alongside recent research, which has identified ribosomes as frequent targets for ubiquitin-mediated degradation during oxidative stress [237], this further points to the precise balance between proteostasis, ribosomes, and the oxidative stress response [238]. Similar changes in cysteine oxidation among chaperones were found in *C. elegans* during treatment with H_2_O_2_, alongside cysteine oxidation in multiple ribosomal proteins [239]. Despite not addressing aging conditions specifically, there are several parallels between these pathways and those identified in the context of different forms of aging (e.g., translation initiation as also seen in Xiao et al. [132]).

Additional studies have found diverse links between specific thiol oxidation and protein homeostasis across numerous other organisms. In plants, for example, many different redox switches have been identified as playing a role in ROS homeostasis, whether in chloroplasts or stroma [240]. Interestingly, studies have shown that cysteine oxidation in *Drosophila melanogaster* does not follow aging, rather that cysteines undergo clear oxidation following fasting [241]. This comes alongside drosophila oocytes, which undergo changes in cysteine oxidation during embryonic development [242], leaving open the possibility of another as-of-yet unknown mechanism of aging- or development-associated oxidation, as well as casting an interesting light on existing aging-associated oxidation changes in mammals, yeast, and plants.

It is important to note that the vastly different experimental setups across various studies may complicate our understanding of the relationship between aging, protein localization, and cysteine oxidation. Varying H_2_O_2_ concentrations are of particular relevance, as these may follow differences in reagent stability and sensitivity. More focused, rigorous studies for each individual protein or pathway discussed above would be required to reconcile the effects measured at different concentrations of H_2_O_2_, different mediums, and other factors. Nonetheless, the range of cysteine modifications identified during aging, caloric restriction, and mild oxidative stress in different organisms provide extensive insights into redox-sensitive cysteines, which may yet emerge as thiol-based switches or functional sensors.

## 10. Cell Cycle and Redox Status Are Highly Connected

In the same way that aging is tightly linked to proteostasis at large, so too is the cell cycle. Another of the most important and well-regulated cellular processes, the cell cycle encompasses a wide range of genetic and proteomic changes during different cellular stages. The link between redox, proteostasis, and the cell cycle is most notable when examining disease models such as cancer, where changes in the cell cycle can lead to catastrophic organism-level damage.

Numerous recent links have been identified between the cell cycle and different forms of redox regulation, whether in the form of individual cysteine redox regulation of checkpoint proteins, such as the human Cdc25C [146] (Figure 3) or the role redox-regulating proteins such as glutaredoxin (i.e., Grx1) may play in activating DNA damage repair pathways [243]. Cell proliferation in particular has been well reviewed as regulated through oxidation [244,245], with accumulation of ROS within the cell ultimately leading to oxidative modifications of numerous cysteines in different cell cycle regulators. Oxidation for the most part leads to inhibition of the individual pathways, though it may on occasion activate proliferation. More recent studies have also shown that this regulation extends to the embryonic level, with ROS levels fluctuating throughout the cell cycle and suggesting cyclic cysteine oxidation during embryonic development [246]. These ROS fluctuations have also been identified at different stages of the cell cycle in human cell lines, with an increase in oxidative damage during mitosis in particular [247]. Taken together, these studies demonstrate the ways in which oxidation—and indeed specific cysteine oxidation—regulate and are regulated by the cell cycle. This has particularly interesting implications that relate to aging and aging-associated disorders, especially in light of cases where single cysteines may regulate folding and function of tumor suppressors (e.g., p16^INK4A^) [248]. Furthermore, numerous studies have identified links between cell division, replicative aging, and oxidation [140,249], finding that increased cell division events correlate with higher ROS levels and global cellular oxidation. Many questions remain as to the mechanisms behind these relationships and present a particularly interesting avenue for future research.

## 11. Conclusions and Perspectives

In this review, we have discussed the intersections between cysteine thiol switches, redox regulation, and protein homeostasis. Thiol switches can be found across a range of proteostasis-associated pathways, with molecular mechanisms that vary widely. These include cysteine modifications on both chaperones and aggregation-prone substrates, as well as redox sensitive cysteines at the heart of the degradation machinery itself. The diversity in modifications and mechanisms—some protective, others harmful—point to robustness in the role cysteine thiols may play within the cell, particularly in these regulatory or sensing roles. Rather than viewing cysteines (and subsequent modifications) as homogenous in either reactivity, chemical mechanism, or “damaging” effect, we find that cysteine modifications appear in many configurations and form a rich tapestry of different molecular mechanisms.

Meanwhile, advancements in redox proteomics and genomic screens as well as in assessing the global redox status have opened the door for further study of the role individual cysteine thiols play in regulating proteostasis. Thus, various studies over the past decade have identified a remarkable range of proteins (many of them explicit chaperones or co-chaperones), which may be additional candidates for redox regulation of proteostasis at large. Individual mechanisms for many such cysteines remain to be studied; however, as we have discussed above, an increasing volume of research into protein folding, localization, and degradation hinges on specific cysteine thiols across the proteostasis network. This may point to a greater number of yet undiscovered dual-functionality within the members of the protein homeostasis system mediated by rapid changes in the redox status of specific redox-sensitive cysteines upon oxidative stress conditions.

Moreover, the pathways that currently demonstrate some form of redox regulation of proteostasis are of particular interest in understanding different human diseases. The links between cellular aging and oxidation have been well studied in correlation with neurodegenerative diseases in particular but may also contribute to a broader understanding of “inevitable” cellular dysfunction and its prevention. Similarly, links between replication and changes in redox regulation have been studied concerning embryonic development and cancer [250], yet may provide insights into the changing landscape of the proteostasis network itself. Together, these are crucial for a deeper understanding of the homeostasis mechanisms themselves and potential treatment or inhibition of harmful processes [251].

The findings presented in this review are but the tip of the iceberg. Future advances in redox proteomics and redox biology will likely reveal an extensive network of oxidation dependence in proteostasis and will uncover novel mechanisms for maintaining a “healthy” proteome during an aerobic lifestyle. To best understand and address when and how the proteome function begins to break down, it is of crucial importance to combine large-scale studies alongside investigating specific cysteine modifications in proteostasis. It is, therefore, also tempting to speculate that this fascinating journey into the redox biology of proteostasis will provide us with detailed knowledge of the cellular redox code and will allow us to use it for biotechnological and therapeutic purposes.

## Figures and Tables

**Figure 1 biomolecules-11-00469-f001:**
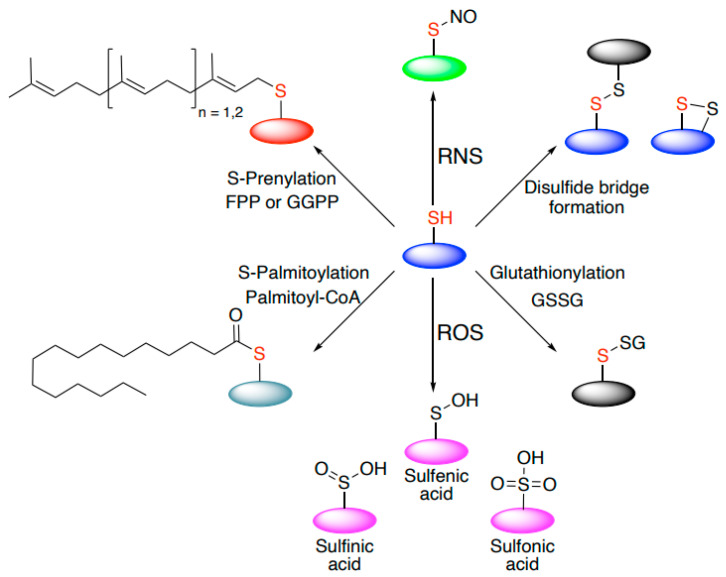
Thiol-based cysteine modifications. Cysteines may undergo a wide range of thiol-based chemical modifications. These include both reversible (e.g., sulfenic acid, disulfide bridge formation, palmitoylation) and irreversible modifications (e.g., sulfinic acid, sulfonic acid, prenylation).

**Figure 2 biomolecules-11-00469-f002:**
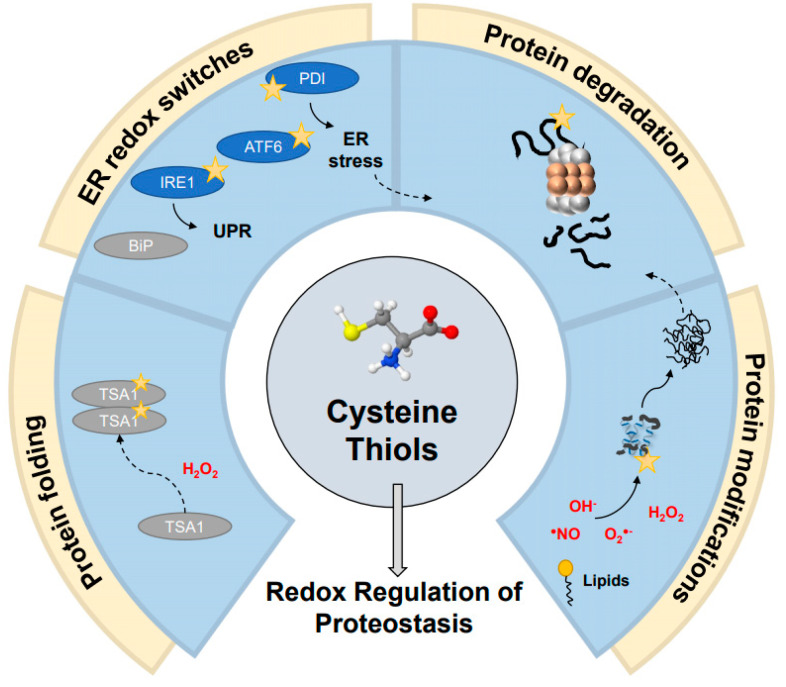
Redox switches in proteostasis. Cysteine thiol switches are involved in regulating various aspects of proteostasis, such as protein folding, ER quality control and the unfolded protein response (UPR), protein degradation across various stages, and protein modifications and maturation. Numerous examples of redox-sensitive thiols have been found across each of these stages of protein quality control, such as redox regulation of the peroxiredoxin TSA1, the ER protein disulfide isomerase (PDI) and BiP chaperones, members of the UPR mechanism IRE1 and ATF6, and the proteasome itself. These emerge from oxidative modifications of varying sorts, including reactive oxygen species (ROS), reactive nitrogen species (RNS), and lipid modifications.

**Figure 3 biomolecules-11-00469-f003:**
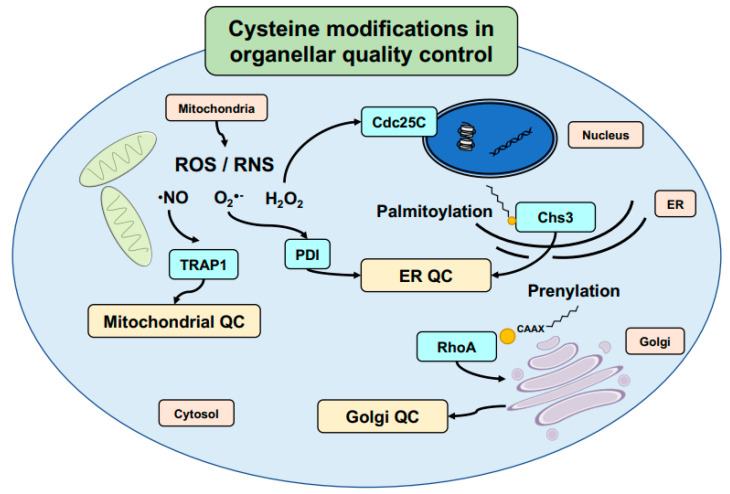
Thiol-modifications define protein folding and translocation. Thiol modifications come in many forms, ranging from reactive oxygen/nitrogen species (ROS/RNS) to lipid modifications during protein maturation. Various individual proteins have been found to have redox-regulated functional changes affecting organellar quality control. Cysteine modifications on chaperones such as TRAP1 and PDI regulate quality control in the mitochondria and ER, respectively, while lipid modifications play additional roles in protein translocation within the cell. Palmitoylation of Chs3 regulates its proper exit from the ER and prevents its aggregation, while prenylation of RhoA alters its interactions with the SmgGDS chaperone. These and other cysteine modifications regulate various additional processes within the cell, such as the cell cycle in the case of hydrogen peroxide regulation of the checkpoint control protein Cdc25C.

**Figure 4 biomolecules-11-00469-f004:**
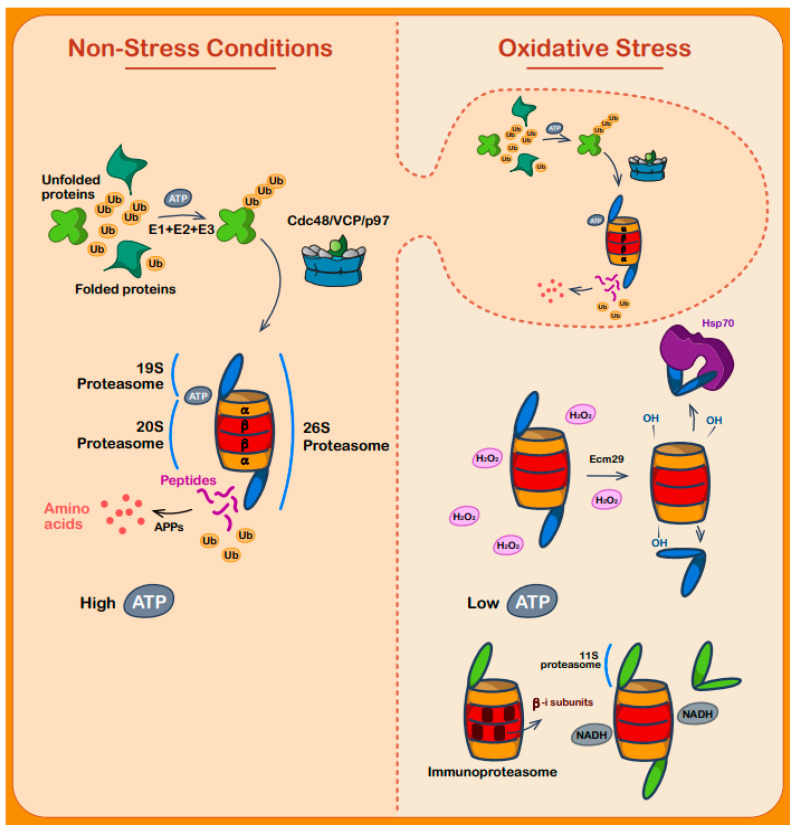
Proteasome-mediated protein degradation during oxidative stress. To the left, under non-stress conditions and in a high ATP environment, unfolded and misfolded proteins undergo degradation by the ubiquitin system by tagging the targeted proteins with ubiquitin (as carried out by the E1, E2, and E3 enzymes), then reaching the 26S proteasome with the aid of different shuttle proteins. The client proteins are initially recognized by the 19S particle. Following binding and utilization of the energy stored in ATP molecules, the substrate unfolds to its primary structure and enters the hollow barrel-like structure of the 20S particle. The substrate is then degraded into peptides by the catalytic units of the beta-subunits, while aminopeptidases (APPs) break them down to amino acids after the peptides exit the proteasome. To the right, oxidative stress conditions alter the cell mechanism of dealing with misfolded and unfolded proteins. The previously described ubiquitin-based degradation system is minimized by the decrease in ATP molecules, and the 26S proteasome comes apart. The 19S and 20S particles split, mediated by the Ecm29 protein and Hsp70. While the 19S is held by Hsp70, the now-oxidized 20S particle begins to function by itself in an ATP-independent manner and degrades unfolded and misfolded proteins. Under these conditions, another kind of proteasome—the 20Si (immunoproteasome)—is upregulated. This proteasome is combined with 3 different beta-subunits (see text), enhancing the proteasome’s catalytic abilities. The 11S or PA28 subunits are upregulated, serving as an alternative regulatory unit for the 20S and 20Si proteasomes, while NADH stabilizes the proteasome structure in the absence of ATP.

**Table 1 biomolecules-11-00469-t001:** Table summarizing select chaperones involved in maintaining protein homeostasis through redox regulation.

Protein Name	Organism(s)	Type	Reactive Cysteine(s)	Activating Oxidant	Additional Information	References
Hsp33	Bacteria *Trypanosoma* *Leishmania*	ATP-independent	232, 234, 265, 268 (*E. coli*)	H_2_O_2_ HOCl		[21,122,123,124]
CnoX	Bacteria	ATP-independent	63 (*E. coli*)	HOCl	Oxidoreductase activity in various bacterial species, not in *E. coli*	[48]
Get3 TRC40/Asna1	Yeast Mammals	ATP-independent	242, 244, 285, 288	H_2_O_2_	When reduced, mediates the delivery of the thiol-anchoring (TA) proteins to the ER	[49]
Ydj1	Yeast	ATP-dependent	185, 188	H_2_O_2_	Part of Hsp40 co-chaperone family	[102,125]
Tsa1 Prx1/Prx2	Yeast, Mammals	ATP-independent	48, 171 (Tsa1) 47 (Prx1/2)	H_2_O_2_	Active as chaperone only when overoxidized	[106,107]

**Table 2 biomolecules-11-00469-t002:** Table of select proteins that contain different cysteine modifications, related to Figure 3. Cysteine modification sites are noted where experimentally verified, alongside general associated processes in which the protein is involved or during which it undergoes the relevant modifications.

Protein Name	Identified Organism	Reactive Cysteine	Modification Type	Associated Protein	References
TRAP1	Human	501	S-nitrosylation	Mitochondrial quality control	[143]
PDI	Human	343	S-nitrosylation	ER quality control	[144]
Chs3	Yeast	Unknown	Palmitoylation	ER protein maturation	[145]
Cdc25C	Human	330, 377	Disulfide bridge formation	Cell cycle checkpoint control	[146]
RhoA	Human	190	Prenylation	Protein–protein interaction	[147]

## Data Availability

Not applicable.

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
