# Peer review of "The Cys Sense: Thiol Redox Switches Mediate Life Cycles of Cellular Proteins"

_biomolecules, 2021, doi:10.3390/biom11030469_

Round 1

Reviewer 1 Report

This review by Radzinski, Oppenheim, and Reichmann focuses on the role of reactive cysteine residues that act as molecular redox switches in response to oxidative stress and proteostasis. The authors discuss various aspects of cysteine sensing in the cell including i) the role cysteine thiols in mediating redox-regulation for proteostasis; 2) cysteine PTMs; 3) thiol editing in the ER by molecular redox switches; 4) protein degradation by redox-sensitive proteins; 5) protein death in response to oxidative stress; 6) correlation between aging and cysteine oxidation; and 7) redox status and cell cycle progression. There are many interesting discussions on aspects of redox chemistry and cysteine thiols in this review, however, many of the passages in the review are confusing and disjointed. Smoother segues between different sections and highlighting the major take-home messages from each section would improve the flow of the manuscript. More background in the introduction on the sources for ROS and RNS production in the cell is also needed. The incorporation of a table to summarize the roles of different chaperones and co-chaperones discussed would improve clarity. The addition of figures to summarize the major cysteine modifications discussed and the chemistry involved/mechanisms involved would also be helpful. All of the acronyms used in the review should also be defined. For the acronyms that are defined, some occur before and others afterward, with some being defined later in the manuscript — including a list of abbreviations used is warranted. The sections focusing on ubiquitylation and the proteasome could be also be improved with more evidence and discussion of the methods used to verify the role of ROS has in regulating E1, E2, and E3 ligase activities. There are many conserved cysteine residues found in members of the E3 ubiquitin ligases, some for zinc coordination (RINGs and RBRs), while others have an absolutely conserved catalytic cysteine required for ubiquitin transfer (RBRs and HECTs). The methods used to verify the 20S regulatory proteins experimentally should be described and discussed. There is also new evidence on Parkin regulation by ROS in the literature that should also be explored. The authors have presented some very interesting insights on cysteine redox and its role in proteostasis, however, the authors need to address these concerns before it can be accepted for publication.

General comments:
There are many comma splices found throughout the abstract and text that disrupt the flow of the manuscript. Many of these could be fixed by substituting “which” with “that”. Please revise.
Hydrochloric acid is HCl, hypochlorous acid is HOCl.
Latin terms should be italicized. i.e. via, Trypanosoma, Leishmania, et al., etc.
RE: Line 259 — since this data has not been peer-reviewed, any reference to unpublished data should not be included in this review.
Figure 1 is informative but is very pixelated. Making some of the images sharper and incorporating sources of ROS and RNS would improve this figure.
Figure 3 is good, but the hand-drawn images could be improved. The authors might want to consider using BioRender to remake this figure. Instead of “abundant” and “scarce”, perhaps the authors could use high [ATP] vs. low [ATP].
Since the attachment of ubiquitin is a posttranslational modification, it should be referred to as “ubiquitylation” and not ubiquitination. This is analogous to other PTMs (i.e. phosphorylation, methylation, SUMOylation, acetylation, etc.)

Minor typos/clarifications that need to be addressed:
Line 92 - stress specificity and
Line 129 - formation or protein misfolding
Line 135 - and accumulation of
Line 154 - the chemical properties
Lines 172-174 - pKa (x3), 9 to 5.7, turning the cysteine thiol in to
Lines 277-278 - Under oxidative conditions, similar to Hsp33, the oxidation of Get3
Line 282 - et al.
Line 353 - such as the cell cycle
Line 375 - (RNS), as previously described in detailed reviews on its
Lines 380-384 - run-on sentence, please revise.
Line 428 - target, it can also serve as
Line 464 - end sentence with “at large.”
Line 495 - 3 “new?” beta-subunits — unclear, please revise.
Line 517-518 - awkward sentence, needs to be revised.
Line 537 - non-stress conditions.
Line 538 - 20S is coordinated by
Lines 547-548 - Please describe the methods used to verify the group of 20S regulatory proteins experimentally
Line 549 - et al.
Line 674 - of the cell cycle
Line 677 - implications that relate to aging
Lines 718-721 - awkward phrasing, please revise.

Reviewer 2 Report

This manuscript presents a survey of the literature in the past 20 years on the relationship between proteostasis and thiol redox regulation. The survey is balanced with respect to the coverage of the contributions from the community and does not over-represent the work from the authors. However, it does not fulfill the main purpose of a review. Namely, to critically integrate the research in the published primary literature towards a synthesis. It is disappointing that although in numerous instances the authors claim that “altogether” findings mentioned in previous paragraphs clarify the redox regulation of some cellular process or have important implications, they then fail to discuss those concrete insights and implications. So, the present survey is a good starting point, but much work remains to be done to turn it into an useful review. In particular, the following shortcomings must be addressed:

  1. Literature prior to 2000 was almost completely overlooked. If the authors’ intention is to review only the contributions in the part 20 years then this needs to be explicitly acknowledged.
  2. There is no evidence that the authors made a critical assessment of the literature, as all the claims appear to have been taken “at face value”, without regard to the physiological pertinence of peroxide concentrations used in many studies, demonstration of the physiological relevance of many Cys modifications, etc.
  3. Many findings are described in vague and generic terms, omitting critical information about model organisms, methodologies and experimental conditions. In some cases the authors present their own generic conclusions about reported findings without clearly explaining how they arrived at them.
  4. Some discussions stray away from the stated focus on the relationship between thiol redox regulation and proteostasis. It would be preferable to keep the focus and discuss the implications of the published research in further depth.
  5. The manuscript would much benefit from the input from someone with a solid understanding of thiol chemistry.

MINOR POINTS

The abstract could be more informative and highlight the main conclusions.

Please define all acronyms — and especially protein ones — on first use.

Consider including a properly referenced table with all the thiol-redox regulated proteins associated to each process discussed in the text, indicating the respective active Cys and the consequences of their oxidation.

Line 76: close parenthesis

Line 92: stress specificity

Lines 146-148 “[…] undesirable thiol oxidation, leading to the addition of negatively charged modifications to the protein in the form of sulfenic or sulfonic acids” This statement needs qualifiers: Protonated thiols are much less reactive with peroxides and other reactive species than thiolates, and in the latter case oxidation does not introduce additional negative charge.

Lines 158: too broad use of the term “redox switches”. Shouldn’t one distinguish between Cys that are directly involved in the catalytic cycle of the enzyme and those that have just a regulatory function.

Line 160: persulfidation

Line 169-170: “through conserved motifs for cysteine – or more specifically thiol – interactions.” Unclear sentence.

Line 171: “their position in the protein sequence”. More correctly, their protein environment, which very often is determined by residues that are distant in the primary sequence.

Line 172 and throughout the text: pKa

Line 173: “five-seven”. Unclear.

Line 198: “reduce cellular peroxide or peroxynitrite derivatives into harmless water molecules” False statement.

Line 209: “The overoxidation of peroxiredoxins is reversible and mediated by sulfiredoxin enzymes” Misleading statement. The overoxidation of peroxiredoxins is not mediated by sulfiredoxin, and it has a large -DG, which makes it quite irreversible. This is why the reaction catalyzed by sulfiredoxin requires the breakdown of one ATP and the oxidation of a reducing equivalent, with a total energetic cost approaching 5 ATP/sulfinic acid.

Line 240: Why “revolutionary”?

Line 256: “its inactivation mechanism is highly conserved and difficult for remodeling”: Unclear sentence. What does “remodeling” mean in this context?

Line 453: What does protein “death” mean? Degradation?

Figure 3:

Characters in the figure are far too small to be easily readable. Please use characters sized at least 7pts.

26S proteasome mislabeled as 20S proteasome in right hand panel.

Line 529: dissociate

Line 538: is coordinated

Lines 542 and following: The operation and significance of the negative feedback loop need to be more clearly explained. Also, how is the mentioned bioinformatics study related to the feedback loop.

Line 690: “The diversity in modifications and mechanisms - some protective, others harmful - point to robustness in the role cysteine thiols may play within the cell, particularly in these regulatory or sensing roles.” Unclear point. Please explain exactly what is meant by “robustness” in this context and how this conclusion was reached.

Reviewer 3 Report

I think that this is a well written review article which I enjoyed reading. I do feel however that some recent concepts of redox signaling could have been addressed. A new redox switching mechanism has been described recently which involves formation of protein persulfides out of sulefnic acids in order to protect proteins from hyperoxidation (Zivanovic et al, Cell Metabolism, 2019). This has been confirmed by another study as well (Doka et al, Sci Advances, 2020). I

Author Response

We thank the reviewer for the positive response and evaluation of our manuscript. We are totally agree with the reviewer suggestion, and the references together with the related sentences were added.

Round 2

Reviewer 1 Report

This revised review by Radzinski, Oppenheim, Methanis, and Reichman is significantly improved from their original submission.  The new additions to the text are comprehensive and through.  The flow of the manuscript is also much smoother and the new/revised figures and tables are very informative and easier to follow. The authors should be commended for making all of the requested changes/edits.  This review will serve as a helpful guide for researchers studying thiol biochemistry in the coming years.

Minor edit:
Cysteine in center of Figure 2 is in the wrong ionic state. Please revise so it is in the zwiterionic form (i.e. +NH3, COO-).

Author Response

We thank the reviewer for the positive evaluation and for pointing out on the  ionic form of Cys in Figure 2. The figure was modified.

Reviewer 2 Report

The present version of the manuscript presents some improvements in organization, style and clarity. However, the main concerns were not substantively addressed. Therefore, the manuscript remains a superficial description of the literature, rather than a review.

Author Response

We thank the reviewer for his time and efforts in reviewing our manuscript.
We are sorry to hear that the revised version didn't meet the expectation. We do believe that this manuscript can contribute to the community, especially to researchers investigating protein quality control, which is the topic of this session.